# Fabrication of Polymethyl Methacrylate (PMMA) Hydrophilic Surfaces Using Combined Offset-Tool-Servo Flycutting and Hot Embossing Methods

**DOI:** 10.3390/polym15234532

**Published:** 2023-11-25

**Authors:** Jianpeng Wang, Yaohui Wang, Guoqing Zhang, Bin Xu, Zejia Zhao, Tengfei Yin

**Affiliations:** 1Guangdong Provincial Key Laboratory of Micro/Nano Optomechatronics Engineering, Shenzhen University, Nan-hai Ave. 3688, Shenzhen 518060, China; 1900291014@email.szu.edu.cn (J.W.); wangyaohui1802@163.com (Y.W.); binxu@szu.edu.cn (B.X.); zhaozejia@szu.edu.cn (Z.Z.); 2Shenzhen Key Laboratory of High Performance Nontraditional Manufacturing, College of Mechatronics and Control Engineering, Shenzhen University, Nan-hai Ave. 3688, Shenzhen 518060, China; 3State Key Laboratory of Ultra-precision Machining Technology, Department of Industrial and Systems Engineering, The Hong Kong Polytechnic University, Hung Hom, Kowloon, Hong Kong, China

**Keywords:** ultra-precision flycutting, hot embossing, microstructure, PMMA, hydrophilic

## Abstract

Polymethyl methacrylate (PMMA) is a material with good surface wettability and has unique and widespread applications in industrial fields. However, fabricating this material in an environmentally friendly way while maintaining its mechanical robustness remains a challenging task. One effective way is through the rational design of microstructure surfaces. The current study fabricated a pyramid microstructure array on a mold surface using offset-tool-servo flycutting, which was then combined with hot embossing to replicate an inverted pyramid microstructure array on a PMMA surface. Firstly, a toolpath compensation algorithm was developed to linearize the arc toolpath and reduce the cost of ultra-precision lathe. Then, the algorithm was further developed to achieve automatic linear toolpath intersection, aiming to ensure the machining accuracy and improve machining efficiency. An experiment testing the linear toolpath intersecting at 90° was conducted, fabricating a pyramid microstructure array with nanoscale roughness on the mold surface. This surface was then employed for replicating an inverted pyramid microstructure array on the PMMA surface using hot embossing. Furthermore, the accuracy of replication was evaluated, and the experimental results demonstrated excellent replication fidelity, exceeding 98%. The microstructural surface of the PMMA exhibited a change in surface wettability. The wettability test showed a water-droplet contact angle reduction from 84.8° ± 0.1° to 56.2° ± 0.1°, demonstrating a good hydrophilic effect. This study introduces a novel, environmentally friendly and high-precision method to fabricate a functional PMMA surface with an inverted pyramid microstructure array. The results of this study also provide strong technical support and theoretical guidance for micro-nanostructure functional surface machining and replicating.

## 1. Introduction

Polymethyl methacrylate (PMMA) has emerged as a promising and representative polymer material due to its unparalleled advantages, including low cost, lightweight characteristics, and recyclability [1,2,3]. Its applications span various fields such as optics [4,5], mechanics [6,7], and biomedical applications [8,9]. However, insufficient hydrophilicity limits its applications in some specific fields, such as coatings [10], sensors [11], and medical devices [12].

Traditional surface modification methods, such as surface deposition [13,14], thermal treatment [15,16], and surface etching [17,18], are typically employed to alter the hydrophilicity of PMMA. However, these chemical processes often require unfavorable working conditions, and they generate undesirable process conditions leading to increased chemical pollution and higher energy consumption [17]. Moreover, the modified PMMA surface is prone to lose its effectiveness in certain extreme environments. Therefore, environmental and durability concerns are driving the industry to seek alternative, safe modification methods.

Microstructural functional surfaces represent an effective solution to the aforementioned challenges. Cai et al. [19] fabricated six microstructures of different sizes on a 7C27Mo2 surface to change the material’s surface wettability. Yang et al. [20] manufactured anti-fogging hydrophilic microstructures on glass by nanosecond laser. Zhou et al. [21] fabricated microstructures with hydrophilicity on a Cf/SiC composite surface via picosecond-laser-induced ablatio. However, the wettability of the material surface is closely related to the quality and arrangement of the surface microstructure. Therefore, it is difficult to study in depth the mapping relationship between the geometric characteristics of the array and the wettability of the material surface for microstructures with low processing quality. In addition, the method of directly fabricating microstructures on the material surface is inefficient and cannot meet production demands.

PMMA, as a thermoplastic material, possesses excellent formation ability. Considering production, processing accuracy, and environmental protection requirements, the current research proposes a combined hot embossing and ultra-precision cutting method for the fabrication of microstructure molds and the production of PMMA functional microstructure surfaces through hot embossing.

Hot embossing has been widely employed in the production of PMMA microstructural surfaces because of its low cost, high efficiency, and high quality, which is beneficial for mass production [22]. Ultra-precision cutting, as a high-quality and high-deterministic physical processing method, is considered a promising method for fabricating large-scale microstructured surfaces and molds [23]. When selecting a microstructure for a PMMA surface, processing cost and efficiency are always pivotal considerations. Different microstructures can indeed influence the wettability of the PMMA surface. However, prioritizing cost-effectiveness in processing led to the utilization of a pyramidal microstructure in the present study. This choice was motivated by the ease with which the pyramid microstructure can be fabricated using an ultra-precision lathe. It circumvents the need for complex boundary conditions or auxiliary equipment, ensuring a high formation speed while only requiring the intersecting of linear toolpaths. This simplified processing method not only helps in cost reduction but also enhances production efficiency. Furthermore, the replicated inverted pyramid microstructure array on the PMMA surface not only significantly alters the wettability of the PMMA surface but also provides superior mechanical robustness compared with a pyramid microstructure array [24,25,26].

Diamond tool flycutting, a form of ultra-precision cutting, removes surface material intermittently and is particularly suited for machining microstructures of straight-groove-type such as pyramid microstructure arrays [27]. Zhu et al. [28] achieved the preparation of a micro-pyramid array with ultra-precision flycutting on the *C*-axis mode. Zhu et al. [29] proposed a machining technique combining flycutting with slow-tool-servo, namely, end-flycutting-servo technology, which generated hierarchical micro/nano structures with pyramid shapes on a four-axis ultra-precision lathe. Jiang et al. [30], aiming to reduce machine costs, first proposed a method for fabricating pyramid microstructures based on linearized cutting with an end flycutting on a three-axis (*X*-, *Z*-, *C*-axis) ultra-precision lathe using a reverse interpolation algorithm to simulate *Y*-axis functionality. The reduction in ultra-precision lathing costs is more advantageous for commodity production. However, their process still necessitates manual intervention and mechanical turntable assistance to intersect the toolpath when fabricating a pyramid microstructure array. This complexity introduces challenges to the flycutting process, consequently diminishing machining precision and efficiency. These limitations to the automated method result in the inability to fabricate inverted pyramid microstructures on the surface of PMMA through the proposed combination approach. This hinders the achievement of high precision and environmentally friendly production, and increases the difficulty of production methods that could improve the wettability of the PMMA surface.

To address these limitations, the current study introduces a new algorithm for the automatic intersection of diamond toolpaths. The proposed algorithm eliminates the need for manual intervention and avoids the use of mechanical turntables. Furthermore, hot embossing experiments were conducted to replicate the inverted pyramid microstructure array on the PMMA surface, and the quality of the inverted pyramid microstructure was measured and analyzed. Finally, water-droplet experiments were performed to validate the hydrophilic properties of the functionalized PMMA surface. The current study presents an environmentally friendly, systemic approach for the production of PMMA hydrophilic surfaces and offers technical guidance and theoretical support for the preparation of high-quality, permanent, multifunctional polymer surfaces.

## 2. Technological Processes

The current research involves the following main technical processes:(1)Developing of a novel algorithm for linearizing arc toolpaths based on the offset-tool-servo flycutting system. This algorithm facilitates automatic intersection of the linear toolpaths at 90°, enhancing the two-dimensional flexibility of the tools and eliminating the need for manual intervention and mechanical turntable assistance. The surface of the mold with a pyramid microstructure array was fabricated using an ultra-precision lathe (Nanotech 450 UPL, Moore, USA) (Figure 1a).(2)Utilizing the self-developed embossing machine (PGM-30, Shenzhen, China) to conduct hot embossing experiments. The P2P (plane to plane) method was employed to replicate an inverted pyramid microstructure array on the PMMA surface (Figure 1b).(3)Measuring the morphology of the microstructures using a white light interferometer (GT-X, Bruker, Germany). The root mean square (RMS) evaluation was performed to assess the quality of the mold surface and the PMMA surface with microstructures (Figure 1c).(4)Conducting water-droplet experiments using a drop shape analyzer (DSA100s, KRUSS, Germany). The hydrophilic properties of the PMMA surface with an inverted pyramid microstructure array were demonstrated (Figure 1d).

## 3. Toolpath Planning

This section introduces a novel algorithm designed to linearize arc toolpaths and implement automatic intersection, thereby enhancing the efficiency, accuracy, and the automation level of microstructure machining on the ultra-precision lathe. Simultaneously, this method contributes to reducing the cost of the lathing process.

There are two key steps when employing the offset-tool-servo flycutting system to fabricate pyramid microstructures on a three-axis ultra-precision lathe:(1)Linearization of the arc toolpath.(2)Generation of linear toolpaths intersecting at 90°.

After the toolpath intersection, the residual part forms the pyramid microstructure.

For clarity in description, we first define the spindle coordinate system as *O_S_-X_S_Y_S_Z_S_* and define the inclination angle *θ* (see Figure 2a) to describe the different straight grooves which intersect in a common center in the plane. Therefore, the range of inclination angles *θ* for straight grooves in the plane is [0°, 180°]. Additionally, to facilitate subsequent modeling calculations, key parameters are defined: *O_d_* represents the center of the diamond tool edge circle, *R_d_* denotes the radius of the diamond tool, *D_p_* is the cutting depth, and the three key points on the interference circle of the diamond tool with the workpiece are labeled *a*_1_, *b*_1_, and *c*_1_, respectively (see Figure 2b). The interference length, representing the straight-groove width, between the diamond tool and the workpiece, is defined as *W_e_*.

### 3.1. Toolpath Planning for 90° Straight Grooves

The process of linearizing the arc toolpath involves subdividing it into infinitesimal microelements and using reverse interpolation to compensate the toolpath within each microelement.

The innovative feature of the current study is the adoption of equidistant (in the *Y*-direction) microelement subdivision, and this method demonstrates algorithmic advantages in the fabrication of straight grooves with varying *θ*. By defining the element length as *h* and the tool’s offset distance as *R*, these two parameters can be employed to output the tool’s rotation angle ∆*c* and compensation distance ∆*x* within each microelement.

Figure 3 illustrates the application of the equidistant (*Y*-direction) subdivision to the offset-tool-servo flycutting system. Here, *O_S_-X_S_Y_S_Z_S_* serves as the spindle coordinate system primarily utilized for describing tool movement since the tool is mounted on the spindle. Meanwhile, *O_W_-X_W_Y_W_Z_W_* is the workpiece coordinate system, which is primarily used to describe the morphology of the straight grooves as they form on the workpiece. Figure 3a shows the entire process of compensating the arc path into a linear path. After compensation, the toolpath changes from the initial arc path *bd* to the linear path *bcd*. However, owing to the different compensation directions of the diamond tool in the *X*-axis during machining, the entire machining process is defined as two parts, including the right compensation stage (*bc*) and the left compensation stage (*cd*), where *bc* equals *cd*. The details are shown in Figure 4b,d, respectively. Figure 4c represents the end of the right compensation stage and the start of the left compensation stage.

In the right compensation stage, *bc* is divided into *n* equal microelements, as shown in Figure 4b. If the microelements are counted with *i* (*I* ∈ [1, *n*]), the height of microelements is represented as *y*_1_ = *y*_2_ = *···y_i_··· = y_n_ = h*. Therefore, for each microelement, the horizontal compensation distance $\Delta x_{bc,i}^{90^{\circ }}$, vertical compensation distance $\Delta y_{bc,i}^{90^{\circ }}$, and rotation angle $\Delta c_{bc,i}^{90^{\circ }}$ can be represented by:

When *i* = 1,
(1)$$\{\begin{array}{l} \Delta y_{bc,1}^{90^{\circ }}=h\\ c_{1}=\Delta c_{bc,1}^{90^{\circ }}=\arccos((R-h)/R)\\ \Delta x_{bc,1}^{90^{\circ }}=\sqrt{{\left[2R\sin(\arccos(((R-h)/R)/2))\right]}^{2}-h^{2}} \end{array}$$

When *i* > 1,
(2)$$\{\begin{array}{l} \begin{array}{l} \Delta y_{bc,i}^{90^{\circ }}=h\\ c_{i}=\arccos((R-h_{i})/R) \end{array}\\ \Delta c_{bc,i}^{90^{\circ }}=c_{i}-c_{i-1}\\ \Delta x_{bc,i}^{90^{\circ }}=\sqrt{{\left[2R\sin((c_{i}-c_{i-1})/2)\right]}^{2}-h^{2}} \end{array}$$

In Equations (1) and (2), the superscript 90° denotes a 90° straight groove, while the subscript *bc* represents the *bc* segment (see Figure 3a). Utilizing the iterative method, the horizontal and vertical compensation distances, as well as the rotation angles obtained in the previous compensation process, are employed in the subsequent compensation. This process is repeated to ascertain the subsequent *X*-axis compensation distance and *C*-axis rotation angle, until the cutting range [1, *n*] is completed.

Similarly, in the left compensation stage, *cd* is divided into *n* equal microelements, as shown in Figure 4d. The microelements are counted with *j* (*j* ∈ [*n* + 1, 2*n*]). Then, *y_n+1_* = *y_n+2_* = *···y_j_··· = y_2n_ = h*. Since the groove compensation processes for the *bc* and *cd* parts are symmetric about the *X*-axis, for each microelement, the horizontal compensation distance $\Delta x_{cd,j}^{90^{\circ }}$, vertical compensation distance $\Delta y_{cd,j}^{90^{\circ }}$, and rotation angle $\Delta c_{cd,j}^{90^{\circ }}$ could be given by:(3)$$\left\{ \begin{array}{l} \Delta y_{cd,j}^{90^{\circ }}=h\\ \Delta c_{cd,j}^{90^{\circ }}=\Delta c_{bc,i}^{90^{\circ }}\\ \Delta x_{cd,j}^{90^{\circ }}=\Delta x_{bc,i}^{90^{\circ }} \end{array}\right.$$
where *i* + *j* = 2*n* + 1.

In Equation (3), the subscript *cd* denotes the *cd* segment (see Figure 3a). Similarly, the calculated compensation distance will be used in the next compensation to determine the *X*-axis compensation distance and *C*-axis rotation angle for the cutting range [*n* + 1, 2*n*], until it is completed.

### 3.2. Toolpath Planning for Inclined Straight Grooves

The toolpath planning model for inclined straight grooves can be derived using the 90° straight-groove toolpath planning model, which demonstrates the advantage of the equidistant microelement (in the *Y*-direction) subdivision method.

The right-inclined straight groove is defined as *θ* ∈ (90°, 180°). To describe the method in detail, we take the representative example of a 135° straight groove, for which the toolpath linearization and microelementation process are illustrated in Figure 4. Essentially, a right-inclined straight groove in the plane is an extension of the 90° straight groove, with the additional horizontal compensation distance *X_c_* for each microelement. The *X_c_* value can be calculated by solving right-angled triangles: $X_{c}=\left|h/\tan\theta \right|$ (see Figure 4b).

Under the conditions of equal division and with counting parameters (*n*, *i* and *j*) identical to the 90° straight groove, and based on Equations (1) and (2), the horizontal compensation distance $\Delta x_{bc,i}^{135^{\circ }}$, vertical compensation distance $\Delta y_{bc,i}^{135^{\circ }}$, and rotation angle within each microelement $\Delta c_{bc,i}^{135^{\circ }}$ of the 135° straight groove’s segment *bc* can be expressed as:(4)$$\{\begin{array}{l} \Delta y_{bc,i}^{135^{\circ }}=h\\ \Delta c_{bc,i}^{135^{\circ }}=\Delta c_{bc,i}^{90^{\circ }}\\ \Delta x_{bc,i}^{135^{\circ }}=\Delta x_{bc,i}^{90^{\circ }}-h/\tan135^{\circ } \end{array}$$

In Equation (4), the superscript 135° represents the 135° straight groove.

Based on Equations (3) and (4), the horizontal compensation distance $\Delta x_{bc,j}^{135^{\circ }}$, vertical compensation distance $\Delta y_{bc,j}^{135^{\circ }}$, and rotation angle within each microelement $\Delta c_{bc,j}^{135^{\circ }}$ of the 135° straight groove’s segment *cd* can be given by:(5)$$\{\begin{array}{l} \Delta y_{cd,j}^{135^{\circ }}=h\\ \Delta c_{cd,j}^{135^{\circ }}=\Delta c_{bc,i}^{135^{\circ }}\\ \Delta x_{cd,j}^{135^{\circ }}=\Delta x_{bc,i}^{135^{\circ }} \end{array}$$
where *i* + *j* = 2*n* + 1.

Similar to establishing the model for right-inclined groove angle in the plane, the left-inclined straight groove is defined as *θ* ∈ (0°, 90°). A left-inclined straight groove in the plane essentially involves subtracting an additional horizontal compensation distance *X_c_* for each microelement compared with that of 90° straight groove. To describe the method in detail, a 45° straight groove is taken as a representative example, for which the toolpath linearization and microelementation process are illustrated in Figure 5.

Similarly, the horizontal compensation distance $\Delta x_{bc,i}^{45^{\circ }}$, vertical compensation distance $\Delta y_{bc,i}^{45^{\circ }}$, and rotation angle $\Delta c_{bc,i}^{45^{\circ }}$ within each microelement of the 45° straight groove’s segment *bc* can be expressed as:(6)$$\{\begin{array}{l} \Delta y_{bc,i}^{45^{\circ }}=h\\ \Delta c_{bc,i}^{45^{\circ }}=\Delta c_{bc,i}^{90^{\circ }}\\ \Delta x_{bc,i}^{45^{\circ }}=\Delta x_{bc,i}^{90^{\circ }}-h/\tan45^{\circ } \end{array}$$

In Equation (6), the superscript 45° represents the 45° straight groove.

According to Equations (3) and (6), the horizontal compensation distance $\Delta x_{cd,j}^{45^{\circ }}$, vertical compensation distance $\Delta y_{cd,j}^{45^{\circ }}$, and rotation angle within each microelement $\Delta c_{cd,j}^{45^{\circ }}$ of the 45° straight groove’s segment *cd* is given by:(7)$$\{\begin{array}{l} \Delta y_{cd,j}^{45^{\circ }}=h\\ \Delta c_{cd,j}^{45^{\circ }}=\Delta c_{bc,i}^{45^{\circ }}\\ \Delta x_{cd,j}^{45^{\circ }}=\Delta x_{bc,i}^{45^{\circ }} \end{array}$$
where *i + j = 2n +* 1.

Based on the preceding analysis, when the *bc* segment is divided into *n* equal microelements, with any part counted by *i*, then the horizontal compensation distance $\Delta x_{bc,i}^{\theta }$, vertical compensation distance $\Delta y_{bc,i}^{\theta }$, and rotation angle $\Delta c_{bc,i}^{\theta }$ within each microelement of the inclined straight groove in the plane can be derived as:(8)$$\{\begin{array}{l} \Delta y_{bc,i}^{\theta }=h\\ \Delta c_{bc,i}^{\theta }=\Delta c_{bc,i}^{90^{\circ }}\\ \Delta x_{bc,i}^{\theta }=\Delta x_{bc,i}^{90^{\circ }}-h/\tan\theta \end{array}$$

Similarly, the *cd* segment is divided into *n* equal microelements, with any part counted by *j*. Then, the horizontal compensation distance $\Delta x_{cd,j}^{\theta }$, vertical compensation distance $\Delta y_{cd,j}^{\theta }$, and rotation angle $\Delta c_{cd,j}^{\theta }$ within each microelement of the inclined straight groove in the plane can be expressed as:(9)$$\{\begin{array}{l} \Delta y_{cd,j}^{\theta }=h\\ \Delta c_{cd,j}^{\theta }=\Delta c_{bc,i}^{90^{\circ }}\\ \Delta x_{cd,j}^{\theta }=\Delta x_{bc,i}^{90^{\circ }}-h/\tan\theta \end{array}$$

Equations (8) and (9) represent mathematical models developed to achieve the linearization of arc toolpaths for straight grooves at various inclination angles, aiming to realize automatic intersecting of linear toolpaths to fabricate straight-groove-type microstructure arrays.

## 4. Experiment and Analysis

### 4.1. Pyramid Microstructures Fabrication and Evaluation

The experiment to fabricate the straight groove was conducted on a three-axis (*X*-, *Z*-, and *C*-axis) CNC ultra-precision lathe. The offset-tool-servo flycutting system incorporates ultra-precision flycutting and slow-tool-servo technology, and the hardware configuration is shown in Figure 6. The tool fixture employed in flycutting (illustrated in Figure 6a,b) was attached to the spindle using a vacuum chuck. Additionally, considering that a curved surface is beneficial for hydrophilicity of materials, an arc radius (*R_d_*) of 274.9 μm (see Figure 6a,d) was employed in the experiment. The diamond tool was mounted on a slot with a rotating radius (*R*) of 15 mm (see Figure 6b), and it followed the rotary and translational servo movements of the flycutting fixture. Finally, considering that ferrous metals can cause severe tool wear, affecting the precision of the pyramid microstructure fabrication, the current study used brass as the workpiece material. The workpiece (see Figure 6a,c) was mounted on a fixture with bolts and followed a translational servo motion along the lathe’s *Z*-axis.

Other relevant information, such as the diamond tool details and machining parameters for the pyramid microstructure, is listed in Table 1.

Based on the toolpath planning model for inclined straight grooves, MATLAB^®^ was employed to calculate a point cloud that could be performed by the ultra-precision lathe, followed by the implementation of linear toolpath intersecting at 90°. After the toolpath intersection, the residual part formed the pyramid microstructure.

The experimental results from the fabrication of the pyramid microstructure array are shown in Figure 7. Figure 7a illustrates the macroscopic surface of the pyramid microstructure array, formed by linear toolpath intersecting at 90° from two directions, creating 2401 pyramid microstructures within a 9.5 mm × 9.5 mm area. Scanning Electron Microscope (SEM, Quanta450FEG, FEI, UAS ) imaging of area A in Figure 7a is shown in Figure 7b, revealing clear geometric shapes of the pyramid microstructure array with uniform square features at the base.

Figure 7c–g are magnifications of five regions in Figure 7b, each containing four pyramid microstructures. The central region exhibits higher quality formation with sharp geometric features at the top (see Figure 7e), while the peripheral regions show pyramid tops with straight-line features (see Figure 7c,g). The region with features at the tops of the pyramid microstructures is indicative of machining errors caused by tool deflection during the rotary processing.

To evaluate the fabrication accuracy, a white light interferometer (Bruker GT-X) was employed to capture the three-dimensional morphology of the fabricated pyramid microstructure array, as shown in Figure 8. Given that the pyramid microstructures are designed with a square base, the analysis focuses on the profile of one side. Figure 8b,c present partial profile curves A_1_-A_1_ and A_1_′-A_1_′, comparing the theoretical and experimental profile curves. Observing the results, the simulated width of the pyramid microstructure array is 195.68 μm, while the measured value is 194.41 μm, achieving replication fidelity of 99.35%. The simulated depth of the pyramid microstructure array is 18 μm, and the experimental value is 18.01 μm, with replication fidelity approaching 100%. The comparison of simulated and experimental values for the width and depth of the pyramid microstructure array indicates good consistency. However, the replication fidelity is influenced by tool setting errors, lathe errors, and material rebound; therefore, these excellent experimental results could not be easily replicated.

To further analyze the accuracy of the pyramid microstructure array, in Figure 8d, the profile measurement curve is fitted to the simulated profile curve to derive a shape error. The *RMS* is introduced to assess the curve error between the two profile curves.
(10)$$RMS=\sqrt{\frac{\sum_{1}^{N}{\left(h_{Exp}-h_{Des}\right)}^{2}}{N}}$$
where *h_Exp_* is the measured height and *h_Des_* is simulation height, and *N* is the number of comparisons.

The calculated *RMS* value is 0.7340 μm. In addition to the boundary regions (as seen in regions B1, B2, and B3 in Figure 8d) which exhibit a larger error attributed to material rebound, the error also increases abruptly due to burrs (as seen in region B4 in Figure 8d). Despite these localized variations, the overall error curve remains relatively stable.

During the linearization process of the toolpath, it is important to highlight that the cutting-edge length (*We*), as shown in Figure 2b, undergoes changes due to deflection. This leads to the transformation of the pyramid microstructure, which transitions from a peak top to a straight top as the microstructure moves away from the center of the array. In Figure 9, a comparison is made between the profile curve of a straight-top pyramid in Figure 7g and a simulated profile curve. The data reveals that the length of the straight-top is 18.1 μm, and the morphology of the pyramid structure experiences significant deformation, with the processing error RMS value increasing to 1.7581 μm. Given the inevitability of this processing error in the current system, mitigation strategies include increasing the flycutting radius (*Rd*) or selectively applying the microstructure array in the central region to reduce the impact of processing errors.

In addition, there are two potential points for improvement in the current algorithm to enhance the accuracy of microstructure fabrication and broaden its applicability to different mold materials. (1) Establishing a mapping relation between the number of discretization points of the toolpath and the quality of microstructure fabrication. In the current research, the linear tool path was discretized into 5000 points. In future experiments, the use of more and fewer discretization points will be explored to find the optimal balance between processing quality and efficiency. (2) Modeling of the mapping relation between flycutting speed and tool wear suppression. In future research, mold materials with better performance will be used, such as mold steel. However, using mold steel necessitates considering the wear suppression of diamond tools. Reducing cutting speed to suppress cutting heat and thereby mitigating catastrophic wear of diamond tools is a novel and appropriate method that can be employed in offset-tool-servo flycutting.

### 4.2. Inverted Pyramid Microstructures Fabrication and Evaluation

The experiment employs hot embossing equipment for P2P-type hot embossing experiments. The entire machine integrates five subsystems, including vacuum heating, nitrogen filling, cold water cooling, force feedback, and control systems to ensure the accuracy of the process (see Figure 10a).

The hot embossing temperature is determined by the glass transition temperature *T_g_* of PMMA. When the temperature is above the *T_g_*, the PMMA is in a soft state which has good stability and molding ability. Differential Scanning Calorimetry (DSC) tests were employed on the PMMA to obtain the *T_g_* (see Figure 10b), which is 112.9 °C. Therefore, the hot embossing temperature needs to be higher than the *T_g_*, and 115 °C was used in the experiment. Additionally, due to the relatively low yield strength of the brass and its susceptibility to deformation with sharp peaks, a relatively small hot embossing pressure of 0.5 kN was selected.

It is essential to note that the current experiment does not incorporate the optimization of hot embossing parameters. Therefore, some experimental parameters were selected based on empirical values. The heating time was determined empirically at 2 min, and the insulation and pressure holding time were contingent on the system cooling to 80 °C. Typically, good hot embossing results for PMMA can be achieved with these empirical values. The specific hot embossing parameters are shown in Table 2.

Figure 11a presents the three-dimensional morphology of the measured inverted pyramid microstructure array obtained through hot embossing replication. Figure 11b, similar to Figure 8b, illustrates the measured surface morphology of the pyramid microstructure array on the brass surface. The measurement results indicate a high and uniform consistency between the inverted pyramid microstructure and the pyramid microstructure.

Detailed analyses of the profile curves A_2_-A_2_ and A_2_’-A_2_’ are shown on Figure 11a and Figure 11b, respectively. The comparative results reveal that the inverted pyramid microstructure faithfully replicates the surface outline of the pyramid, maintaining equal spacing between adjacent microgrooves as anticipated. The difference between the measured depth and the design depth is less than 1 μm. Specifically, the inverted pyramid microstructure exhibits a width of 193 μm and a height of 17.74 μm. Their replication fidelity values are 99.27% and 98.50%, respectively, compared with the pyramid microstructure. Furthermore, the surface curve of the inverted pyramid microstructure is relatively smooth, indicating fewer burr defects and less noticeable material rebound.

For the convenience of error evaluation, Figure 11c is mirrored and then compared with Figure 11d for error calculation, as shown in Figure 11e. The *RMS* value of the curve error is 0.8162 μm, indicating a good fit between the two profile curves. Fluctuations in the measured profile curve may be attributed to tool wear or burrs and the pulses on the error curve may be caused by burrs formed during hot embossing and flycutting processes or material deformation. Additionally, the larger deviations in the error curve are concentrated near the sharp point, where material compression tends to induce significant deformation.

To provide a clear and detailed view of the surface features, Figure 12 shows the microscopic morphology of the inverted pyramid microstructure array on the PMMA surface and the pyramid microstructure array on the brass surface. The processed structures are arranged uniformly.

Figure 13 presents a comparative analysis between the theoretical and hot embossing inverted pyramid microstructure arrays. In this comparison, Figure 13a illustrates the simulated three-dimensional morphology of the inverted pyramid, while Figure 13b, consistent with Figure 11a, shows the measured three-dimensional morphology of the hot embossing inverted pyramid microstructure array. Profile curves A_3_-A_3_ and A_3_’-A_3_’ represent partial profiles of Figure 13a,b, as shown in Figure 13c,d.

In comparison with the theoretical values, the values for the width and height replication fidelity of the hot embossing inverted pyramid microstructure are 98.63% and 98.55%, respectively, indicating noticeable material rebound (see Figure 13e). Further error analysis is conducted, as shown in Figure 13e, where the RMS value is 0.8123 μm. Similarly, significant deformation is also observed near the sharp point.

The above analysis demonstrates that hot embossing is an effective method for replicating high-quality pyramid microstructures fabricated using the proposed method. The replication fidelity of both the width and height of a single inverted pyramid microstructure is above 98%, compared with the pyramid microstructure and designed values. This high-quality replication fidelity provides corresponding technical support for the mass production of functional microstructured PMMA surfaces.

### 4.3. Wettability Testing

Water-droplet experiments were further conducted on the PMMA surfaces to demonstrate the function of the inverted pyramid microstructure array.

As shown in Figure 14a, the initial PMMA surface presents relatively weak hydrophilicity, with a water contact angle of 84.8° ± 0.1°. In contrast, for the PMMA surface with an inverted pyramid microstructure array, the water contact angle decreases to 56.2° ± 0.1°, representing a good hydrophilicity.

Wenzel [31,32] proposed the classical model that represents the contact angle with the microstructure surface. The contact angle *θ** of rough surfaces has the following relation with the contact angle *θ_c_* of smooth surfaces:(11)$$\theta^{*}=\arccos(r\cos\theta_{c})$$

Here, *r* is the roughness factor of the material surface, representing the ratio of the actual contact area to the intrinsic contact area of the solid/liquid interface. In comparison with smooth surfaces, the microstructure surface of PMMA increases the overall surface area, leading to a greater contact area at the solid/liquid interface, consequently causing an increase in *r*. The intrinsic contact angle *θ_c_* is a constant value, approximately 84.8°. Therefore, as r increases, *θ** decreases accordingly, resulting in the PMMA surface exhibiting enhanced hydrophilicity. In the present study, the calculated *r* is approximately 6.138.

The variation in hydrophilicity observed on the PMMA surface serves as a clear manifestation of the effectiveness of the inverted pyramid microstructure array. Furthermore, the versatility of the approach is evident as modifications to machining parameters (cutting depth and tool radius) offer a means to tailor the dimensions, both in terms of width and depth, of the pyramid microstructures. This not only highlights the adaptability of the fabrication process but also underscores its potential for precisely controlling the wettability properties of the PMMA surface.

## 5. Conclusions

The present study proposes a novel method to fabricate a precision inverted pyramid microstructure array on a PMMA surface, aiming to change the surface wettability of PMMA through the combination of offset-tool-servo flycutting and hot embossing. The environmentally friendly machining process is beneficial due to reduced chemical pollution. The key points can be concluded as follows:A novel arc toolpath linearization algorithm was developed to simulate the *Y*-axis functionality of ultra-precision lathes and reduce the cost of ultra-precision lathing.The automatic intersecting of linear toolpaths was achieved utilizing a three-axis ultra-precision lathe to fabricate a pyramid microstructure array on the mold surface. This enhances the efficiency and precision of microstructure fabrication, advancing the level of automation in ultra-precision lathes.Hot embossing was employed to replicate an inverted pyramid microstructure array on the PMMA surface, achieving precise microstructure formation with a replication fidelity exceeding 98%.The inverted pyramid microstructure array effectively enhanced the hydrophilicity of the PMMA surface, reducing the water contact angle from 84.8° ± 0.1° to 56.2° ± 0.1°.

Future research will explore the algorithm and microstructure optimization, as well as the scalability and commercialization of various mold materials.

## Figures and Tables

**Figure 1 polymers-15-04532-f001:**
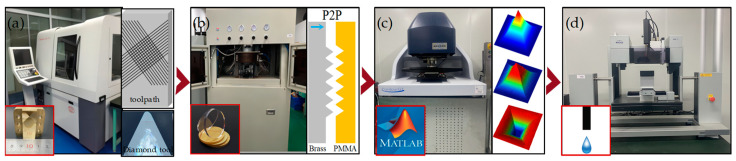
The process flow of PMMA hydrophilic surfaces fabrication. (**a**) Ultra-precision cutting of precision mold with a pyramid microstructure array; (**b**) replicating an inverted pyramid microstructure array on PMMA surface employing hot embossing; (**c**) surface measurement and evaluation of fabricated microstructures; (**d**) wettability testing.

**Figure 2 polymers-15-04532-f002:**
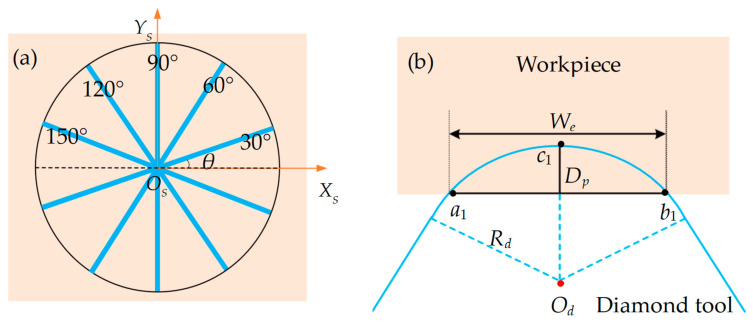
(**a**) Schematic diagram of straight grooves at different inclination angles; (**b**) definition of key parameters for the diamond tool during cutting.

**Figure 3 polymers-15-04532-f003:**
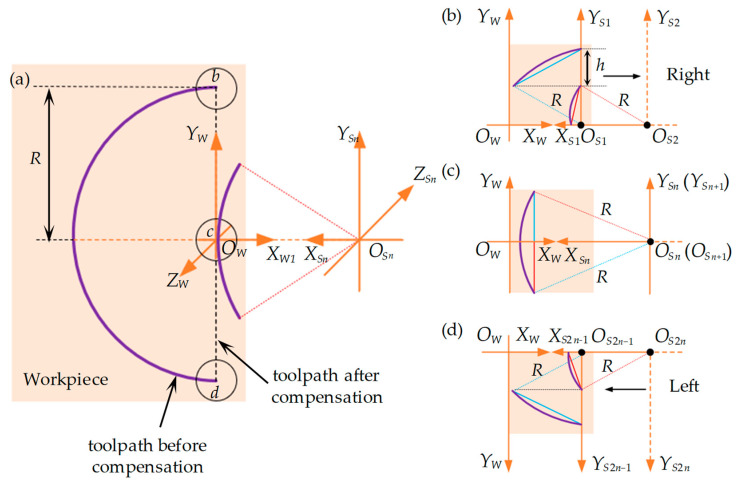
Schematic diagram of toolpath compensation in microelements for 90° straight grooves. (**a**) Comparison of arc toolpath compensation before and after; (**b**) right compensation stage of toolpath in the first two microelements; (**c**) toolpath in the *n* and *n +* 1 microelements; (**d**) left compensation stage of toolpath in the 2*n* − 1 and 2*n* (last two) microelements. The purple line represents the arc tool path before compensation.

**Figure 4 polymers-15-04532-f004:**
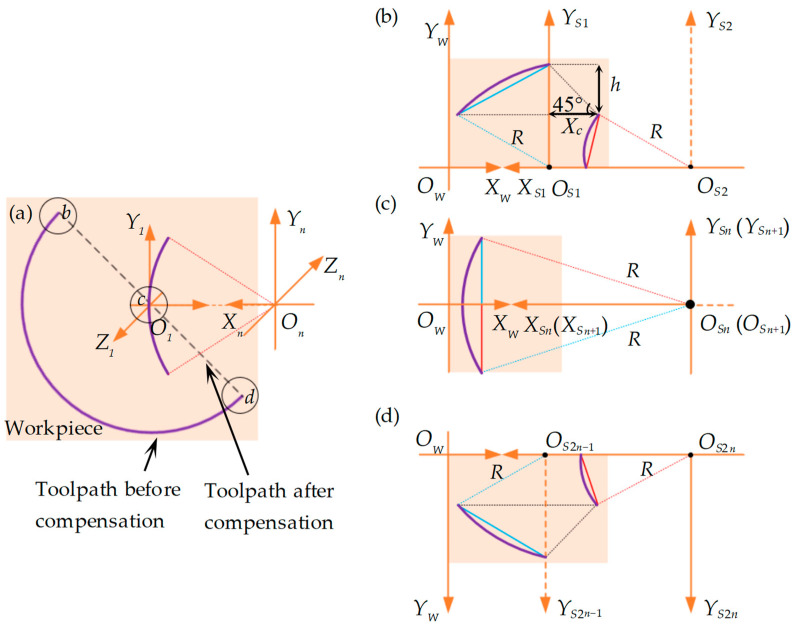
Schematic diagram of toolpath compensation within the microelements of a 135° straight groove. (**a**) Comparison of arc toolpath compensation before and after; (**b**) toolpath compensation of *bc* segment in the first two microelements; (**c**) toolpath in the *n* and *n +* 1 microelements; (**d**) toolpath compensation of *cd* segment in the last two 2*n* − 1 and 2*n* microelements. The purple line represents the arc toolpath before compensation.

**Figure 5 polymers-15-04532-f005:**
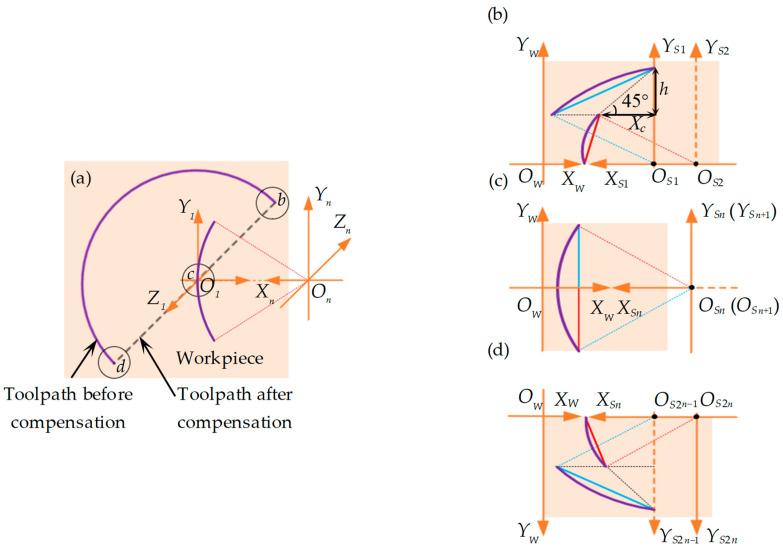
Schematic diagram of the toolpath compensation within the microelements of a 45° straight groove. (**a**) Comparison of arc toolpath compensation before and after; (**b**) toolpath compensation of *bc* segment in the first two microelements; (**c**) toolpath in the *n* and *n +* 1 processed microelements; (**d**) toolpath compensation of *cd* segment in the last two 2*n* − 1 and 2*n* microelements. The purple line represents the arc toolpath before compensation.

**Figure 6 polymers-15-04532-f006:**
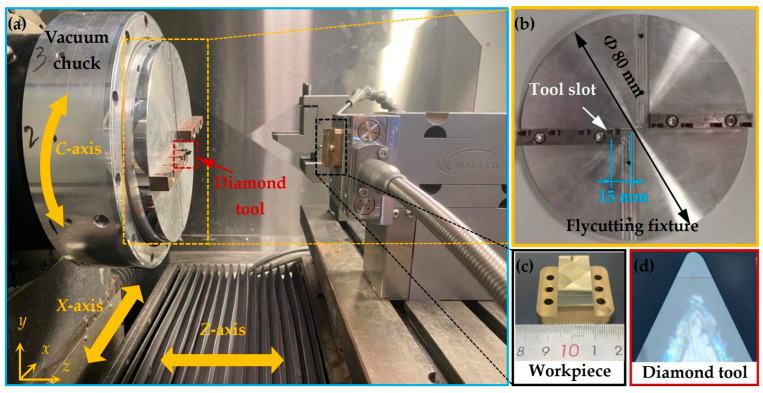
Experimental configuration and details. (**a**) Configuration of the offset-tool-servo flycutting system; (**b**) parameters of the flycutting fixture; (**c**) workpiece; (**d**) diamond tool.

**Figure 7 polymers-15-04532-f007:**
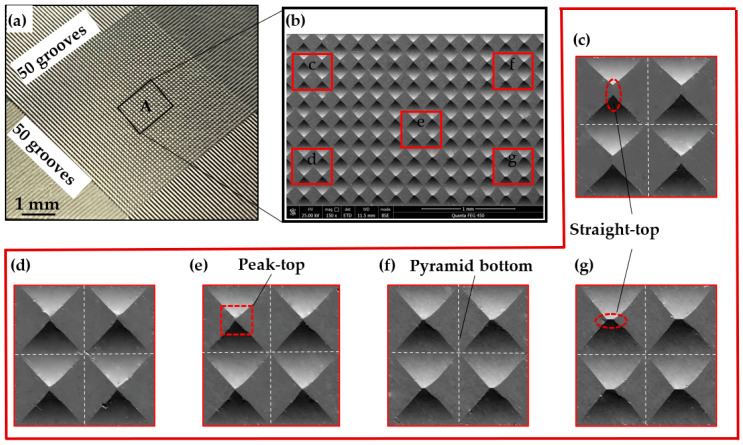
Morphology of a pyramid microstructure array on brass surface. (**a**) Macroscopic representation of the pyramid microstructure array formed by intersecting linear toolpaths; (**b**) SEM captured surface morphology of the pyramid microstructure array; (**c**–**g**) are magnification views of the corresponding regions in (**b**).

**Figure 8 polymers-15-04532-f008:**
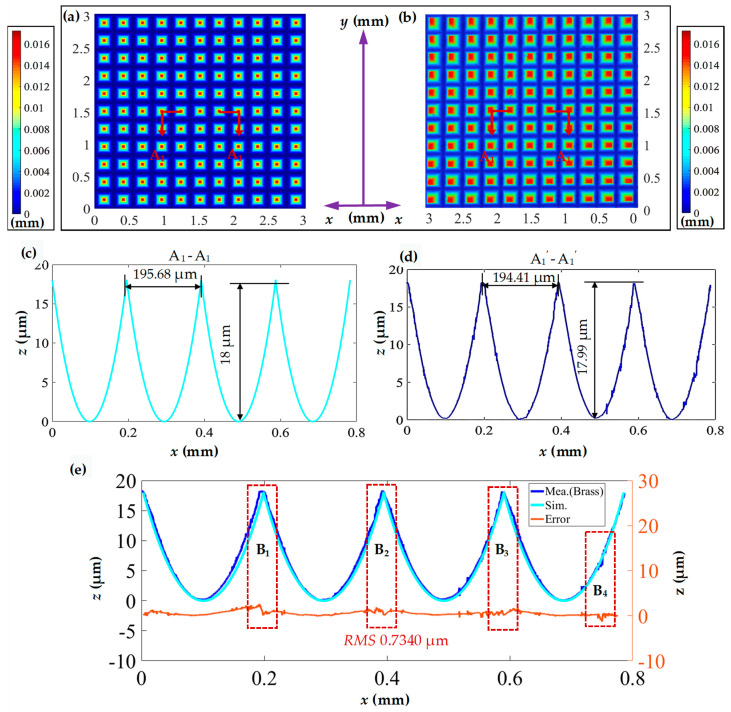
Measured and simulated morphology comparison of the pyramid microstructure array. (**a**) Simulated three-dimensional morphology of the pyramid microstructure array; (**b**) measured three-dimensional morphology of the machined pyramid microstructure array; (**c**,**d**) profile curves in the horizontal direction; (**e**) comparison of profile curves and error analysis.

**Figure 9 polymers-15-04532-f009:**
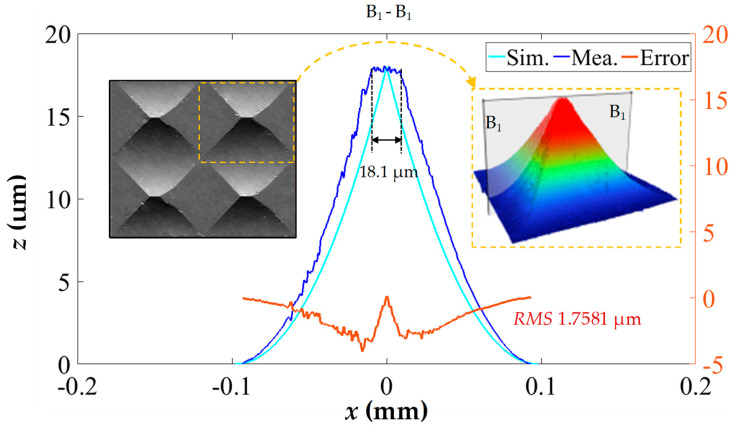
Measured and simulated morphology comparison of a straight-top pyramid microstructure in corner area.

**Figure 10 polymers-15-04532-f010:**
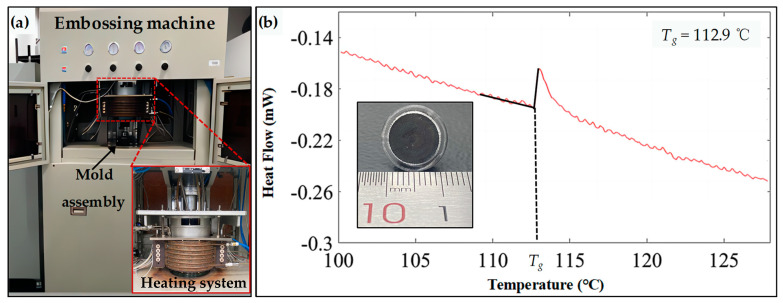
Experimental processing layout. (**a**) Hot embossing machine configuration. (**b**) DSC test for PMMA.

**Figure 11 polymers-15-04532-f011:**
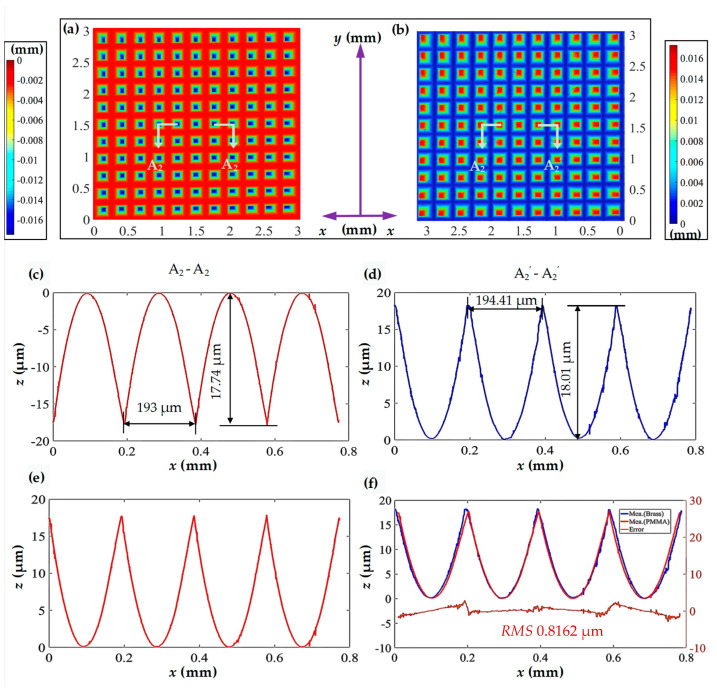
Comparison of the measured results for the pyramid microstructure array and the inverted pyramid microstructure array. (**a**) Measured three-dimensional morphology of the machined pyramid microstructure array; (**b**) measured three-dimensional morphology of the inverted pyramid microstructure array; (**c**,**d**) profile curves in the horizontal direction; (**e**) mirrored curve about the *X*-axis for (**c**); (**f**) comparison of profile curves and error analysis.

**Figure 12 polymers-15-04532-f012:**
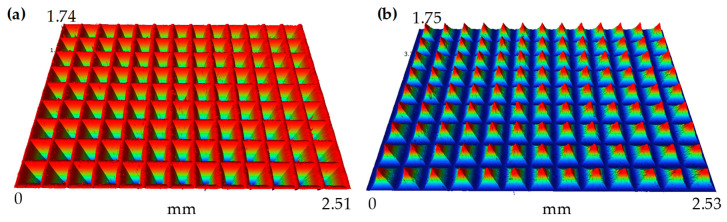
Microphotograph of (**a**) the inverted pyramid microstructure array on the PMMA surface and (**b**) the pyramid microstructure array on the brass surface.

**Figure 13 polymers-15-04532-f013:**
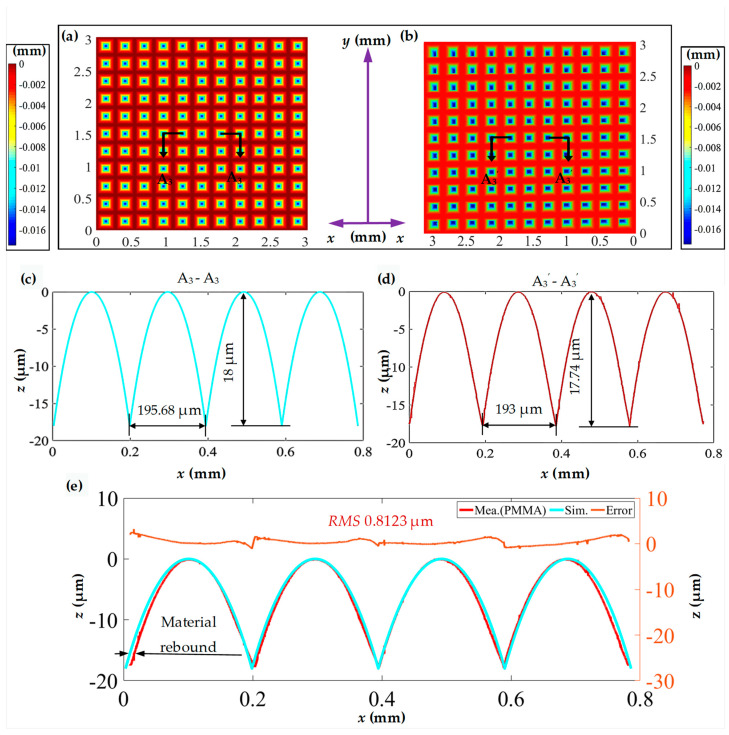
Simulated and measured results comparison for the inverted pyramid microstructure array. (**a**) Simulated three-dimensional morphology of the inverted pyramid microstructure array; (**b**) measured three-dimensional morphology of the pyramid microstructure array; (**c**,**d**) profile curves in the horizontal direction; (**e**) comparison of horizontal profile curves and error analysis.

**Figure 14 polymers-15-04532-f014:**
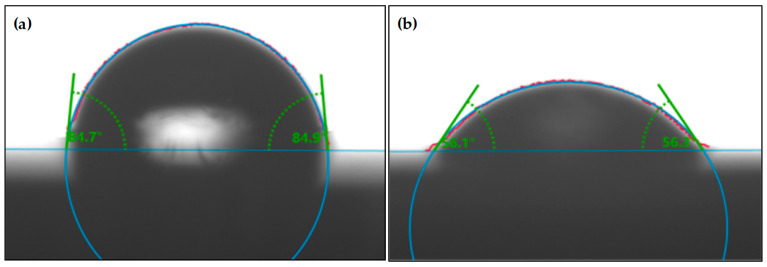
Wettability testing for PMMA surface. (**a**) Initial PMMA surface; (**b**) PMMA surface with an inverted pyramid microstructure array.

**Table 1 polymers-15-04532-t001:** Processing parameters of the pyramid microstructure.

Parameter		Value
Diamond tools	Tool radius *R_d_*	274.9 μm
	Rake angle	0°
	Clearance angle	10°
	Included angle	60°
Ultra-precision lathe	Cutting depth *D_p_*	18 μm
	Number of equal divisions *n*	5000
	Micro-elements length *h*	3 μm
	Number of straight grooves (45°/135°)	50/50
	Intersection angle of straight grooves	90°

**Table 2 polymers-15-04532-t002:** Hot embossing parameters of inverted pyramid microstructure array.

Parameters	Temperature	Pressure	Insulation Time	Pressurization Time
Value	115 °C	0.5 kN	2 min	6 min

## Data Availability

The data presented in this study are available on request from the corresponding author. The data are not publicly available due to privacy.

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
