# Peer review of "Fabrication of Polymethyl Methacrylate (PMMA) Hydrophilic Surfaces Using Combined Offset-Tool-Servo Flycutting and Hot Embossing Methods"

_polymers, 2023, doi:10.3390/polym15234532_

Round 1

Reviewer 1 Report

Comments and Suggestions for Authors

The present manuscript addresses the problem of the elaboration of an effective technique for fabricating a functional microstructure on the PMMA surface in order to modify its properties (wettability and mechanical robustness). The combination of ultra-precision cutting of a brass mold with a pyramid microstructure array and its implementation for replicating of an inverted pyramid texture on the PMMA surface by hot embossing was tested.  An algorithm for linearizing arc toolpaths and implementation of automatic intersection was elaborated for improving the accuracy and affordability of the cutting method. The findings of the study are presented clearly, and usability of the results is obvious.   

I vote for the publication of this paper in the ‘Polymers’ journal and provide recommendations only on adding some optional details to the text.

No

Comment

1

In the Introduction section, an inverted pyramid microstructure array was claimed to be a good solution for improving the PMMA surface wettability and mechanical robustness. Were other microtextures, for instance, with triangular, square, hexagonal, and circular cross-sections, or complex hierarchical structures, considered for potential use?

A brief comparison of the advantages and drawbacks of different patterns (tested for PMMA) would be appreciated.

2

Are there any assumptions on how the dimensions (length/width and depth) of the pattern ‘cells’ may affect the hydrophilic performance of PMMA (or similar materials)? Could it be a monotonic dependence of the wettability on the sizes?

3

What considerations were used for the selection of the pyramid’s sizes? Are there any known established ranges of sizes for PMMA (or similar materials) which were proved to improve hydrophilic properties?

4

If the elaborated algorithm is readjusted for the fabrication of a texture of pyramids with other sizes, do the authors expect any difficulties with maintaining the machining precision?

5

Some ideas on hardware adjustments were mentioned in the text.

If the authors have any ideas on how to compensate for the effects of the machining errors and the imperfections associated with the mold material (brass and, probably, other ones) by adjusting the proposed algorithm (if reasonable), they should not hesitate to include them in the manuscript.

6

In the manuscript, the divergence of the experimental pyramid’s forms from the calculated ones was analyzed for the central section only wherein the pyramids had ‘regular’ shapes. If it is possible, please, include the respective results for at least one of the ‘remote’ zones (for example, near the corners of the microstructure arrays shown in either of Figures No. 8, 10, and 11).

7

Attaching a microphotograph of the textured PMMA surface would be useful as well.

8

Is there a chance to fabricate (in future) a multiple-use mold of a stronger material (for better maintaining of its form after embossing cycles)?

Reviewer 2 Report

Comments and Suggestions for Authors

1. Language revision is required. Avoid using we,..., grammar mistakes.

2.  Referring to the references should be in square brackets.

3. At the end of the introduction, a paragraph for the objective of the current work should be added.

4. References should be mentioned for the tables 3 and 4.

5. Figures 5, 6, and 7 should be presented as histograms to have a picture of many cases of corrosion rates simultaneously.

6. Avoid reasoning in the conclusion section, just mention the concluding remarks based on the results.

The title should be modified as follows:

Mechanical properties and the corrosion rates estimation for the AISI 316 stainless steel under the heat treatment and cold working

Comments on the Quality of English Language

The English language is poor, it needs a proofreader to revise the manuscript completely.

Reviewer 3 Report

Comments and Suggestions for Authors

This manuscript, “Study of PMMA Hydrophilic Surfaces Fabrication through the Synergy of Offset-tool-servo Flycutting and Hot Embossing “, provided an environmental-friendly strategy for modification of surface wettability of PMMA by fabrication of inverted pyramid microstructure array on the surface, which is a good demonstration and guidance for the modification of surface functionality using hot embossing and replicating methods.

The reduction of water contact angle demonstrated the improvement of surface hydrophilicity.

Overall, the experiment is properly designed, the characterization is sufficient for its content, and the conclusions are supported. So, I would like to recommend it to Polymers.

The quality would be greatly improved if there were more embodiments presented in the paper with a series of microstructure arrays and pyramid sizes.

Comments on the Quality of English Language

Overall, the English is fine, while a minor editing of English language is still needed.

Reviewer 4 Report

Comments and Suggestions for Authors

The paper is good and may be revised as follows:

1. Expand relevant literatures with critical findings and mention the research gap.

2. The comparison of simulated and experimental values for the width and depth of the pyramid microstructure array indicates good consistency. Explain with reasons citing with previous literature support.

3.  The comparison of simulated and experimental values for the width and depth of the pyramid microstructure array indicates good consistency. Explain briefly the phenomenon with previous findings.

4. The change of the hydrophilicity on PMMA surface demonstrates the function of the inverted pyramid microstructure array. Justify.

5. Write the future scope of research.

Round 2

Reviewer 3 Report

Comments and Suggestions for Authors

no more comments

Reviewer 4 Report

Comments and Suggestions for Authors

The paper has been revised and thus may be accepted.